# Psychometric properties of the item-reduced version of the comprehensive general parenting questionnaire for caregivers of preschoolers in a Finnish context

Carola Ray[1,2]*, Ester van der Borgh-Sleddens[3], Rejane Augusta de Oliveira Figueiredo[1], Jessica Gubbels[4], Mona Bjelland[5], Eva Roos[1,6,7]

1 Folkhälsan Research Center, Helsinki, Finland, 2 Department of Food and Nutrition, University of Helsinki, Helsinki, Finland, 3 Mondriaan Mental Health Center, Heerlen, The Netherlands, 4 Department of Health Promotion, NUTRIM School of Nutrition and Translational Research in Metabolism, Maastricht University, Maastricht, The Netherlands, 5 Department of Nutrition, University of Oslo, Oslo, Norway, 6 Department of Public Health, University of Helsinki, Helsinki, Finland, 7 Department of Food studies, Nutrition and Dietetics, Uppsala University, Uppsala, Sweden

* carola.ray@folkhalsan.fi

**Data Availability Statement:** Both datasheet files are available from the the public repository OSF

## Abstract

### Introduction

Many instruments for assessing general parenting have been reported as burdensome and are thus seldom used in studies exploring children's energy balance-related behaviors or weight. This study evaluates the factorial structure of the item-reduced version of the Comprehensive General Parenting Questionnaire (CGPQ), which assesses five constructs of general parenting.

### Methods

The study uses data from two cross-sectional studies: Study 1 in 2014 (n = 173) and Study 2 in 2015–16 (n = 805). Parents of children aged three to six answered the CGPQ; in Study 1 the 69-item version, and in Study 2 the 29-item version. The reduction was based on the results of the confirmatory factor analyses (CFA) in Study 1. In both datasets, internal consistency, as Cronbach's alphas and intraclass correlations between the items of each construct, was tested. A combined assessment of the CFA and items response theory evaluated the construct validity and the item importance for the 29-item version, and a further the reduced 22-item version.

### Results

In Study 1, the highest Cronbach's alphas were shown for the five constructs in the 69-item version. A higher intraclass correlation was found between the constructs in the 69- and 29-item versions, than between the 69- and the 22-item version. However, a high concordance was found between the constructs in the 29- and 22-item versions in both Study 1 and in Study 2 (0.76–1.00). Testing the goodness-of-fit of the CFA models revealed that the 22-

Registries, and the DOI is: DOI 10.17605/OSF.IO/
VQN9X.

**Funding:** The DAGIS project and this study was
financially supported by Folkhälsan Research
Center, University of Helsinki, The Ministry of
Education and Culture in Finland (ER), The Ministry
of Social Affairs and Health (ER), The Academy of
Finland (Grant: 285439, 315816) (ER), the Päivikki
and Sakari Sohlberg foundation (ER), and the
Medicinska Föreningen Liv och Hälsa (ER). The
funders had no role in study design, data collection
and analysis, decision to publish, or preparation of
the manuscript.

**Competing interests:** The authors have declared
that no competing interests exist.

item model fulfilled all the criteria, showing that it had a better factorial structure than the 29-item model. Standard estimations ranged from 0.20 to 0.76 in the 22-item version.

## Conclusion

The reduced 22- and 29-item versions of the 69-item CGPQ showed good model fit, the 22-item version the better of the two. These short versions can be used to assess general parenting without overburdening the respondents.

## Introduction

Childhood overweight and obesity remains a significant public health challenge, even though its incidence seems to be decreasing in some countries [1, 2]. A wide variety of obesity-preventing interventions have been reported [3–5]. Environmental factors, both physical and psychosocial, are highly relevant for children's energy balance-related behaviors (EBRBs), which are behaviors related to food habits and physical activity [6]. For young children's EBRBs, factors relating to parents are of importance, as young children are highly dependent on their parents [7, 8].

Previous studies have shown associations between parenting practices that promote healthy EBRBs, and children's actual healthy EBRBs [9–11]. Such parenting practices are related to specific situations and contexts [12]. A more general approach of parenting, such as the emotional climate in which parents socialize their child and their attitudes towards the child [12], also influences children's EBRBs and weight status, as shown in several studies, but not all [9, 13–17]. In addition, it seems that parenting style might also predict changes in childhood diet [18]. Darling and Steinberg have proposed in their contextual model of parenting style, that parenting style moderates the associations between parenting practices and children's outcomes [12]. This contextual model in which parenting style is a moderator has been brought up in several studies examining the associations between parenting practices and children's EBRBs [8, 19–21]. Although some studies have found that parenting style is moderating the association between parenting practices and children's and adolescent's food intake or BMI [22–26], to the best of our knowledge, only one study has examined young children (one- to three-year-olds) [27].

Traditionally, general parenting has been assessed by two dimensions; warmth and control [28]. Crossing these dimensions results in four parenting styles; authoritative (high in warmth, high in control), authoritarian (low in warmth, high in control), permissive (high in warmth, low in control), and uninvolved (low in warmth, low in control) [12, 29]. However, assessing parenting style on the basis of only two dimensions might oversimplify the concept of parenting. Therefore, Sleddens and colleagues developed a new instrument, including the construct of structure and simultaneously splitting the construct of control into behavioral control, coercive control and overprotection [30]. By splitting control into three constructs, the instrument enables a distinction between positive (behavioral control) and negative control (coercive control and overprotection). In addition, overprotection seemed to be less studied in relation to children's EBRBs than the other control concepts, and a valid instrument for examining overprotection, parenting practices, and children's EBRBs was needed [30]. The developed instrument was named the Comprehensive General Parenting Questionnaire (CGPQ), and included 85 items [30]. It was aimed at parents of children aged 5 to 13 years, and it has been used as such to examine associations between the five parenting constructs and children's health

behaviors [31]. Some studies used only the "positive" parenting constructs (nurturance, structure and behavioral control) in examining associations between parenting practices, health behaviors and BMI and the role of general parenting, which meant that the two negative (control constructs (coercive control and overprotection) were unexplored [27, 32]. A version for parents of children aged one and four to five was later developed, which included 69 items [33]. Although somewhat shortened, the 69-item version was still considered too burdensome to be included in comprehensive weight-related child studies, as was too time-consuming and complicated instrument for assessing parenting [34]. Shorter instruments for assessing parenting were therefore urgently needed.

The Increased Health and Wellbeing in Preschools (DAGIS) cross-sectional pilot study in 2014, hereafter Study 1, assessed general parenting using the 69-item CGPQ. Based on the responses in Study 1, the number of items was reduced to 29, but these still included all five original higher-order constructs. This 29-item version of the CGPQ was then used in the next step, in the DAGIS-Survey in 2015–2016, hereafter Study 2 [35, 36]. The aim of this study was to evaluate the factor structure of the reduced 29-item version of the 69-item CGPQ for young children. In addition, the study tested the factorial structure of other possible models with a reduced number of items, and the concordance of the reduced versions with the original version of the 69-item CGPQ.

## Methods

### Participants and data collection

This study used two separate datasets, both collected as part of the Increased Health and Wellbeing in Preschools research project (DAGIS) [36]. First, in October–November 2014, a convenience sample was collected in the DAGIS Pilot study (Study 1) through an online survey for parents of three- to six-year-old children in Finland. The online questionnaire was open for three weeks. Participants were recruited through social network (Facebook, Twitter) websites for parents of young children, Facebook pages of organizations arranging activities for families with young children such as the Mannerheim League for Child Welfare (www.mll.fi), Folkhälsan (www.folkhalsan.fi/en/om-folkhalsan/), and through preschools (emails). The users of several of the recruitment channels were families whose children might not attend preschool full-time (four or more days per week). Parents in Study 1 were asked to go to the DAGIS project website (www.dagis.fi) and to open the link to the online survey. In total, 173 parents completed the online questionnaire. Second, this study used data from the DAGIS Survey study (Study 2), conducted in eight Finnish municipalities from September 2015 to April 2016 [35]. Recruitment was conducted through preschools, and parents of children aged three to six years were invited to participate. The final response rate was 24% and parents of 864 children participated in the study [35].

### Ethical approval

The design and the methods in Study 1 were in line with The Ethical Principles of Research with Human Participants and Ethical Review in the Human Sciences in Finland [37]. The respondents were informed by written instructions about the study before they began the questionnaire, and they were told how the data would be handled and that the responses would be processed anonymously. The respondents were also informed at the beginning of the online questionnaire that by answering they were agreeing to participate in the study. Study 2 was approved as ethically acceptable by the University of Helsinki Ethical review board in humanities and social and behavioral sciences in February 2015 (6/2015). In study 2 the participating parents signed an informed consent before study 2 started.

## Methods and measurements

**The comprehensive general parenting questionnaire.** Both Study 1 and Study 2 study included items derived from the CGPQ, but had a different number of items. Study 1 used the original 69-item version of the questionnaire for children aged between one and four [33]. The 69-item questionnaire was translated into the two official languages of Finland, Finnish and Swedish, even though Study 1 used only the Finnish version. The translation was in line with standard recommendations, and included a back-translation [38]. The developer (EB) of the CGPQ was involved in all stages of the translation. The comprehensibility of the translated questionnaire was pretested on four parents of three- to six-year-old children, through interviews on their interpretation of the questions. Finnish- and Swedish -speaking researchers in the DAGIS team (CR, ER) approved the final versions of the CGPQ.

The 69-item questionnaire assesses five higher-order constructs of parenting; nurturance, structure, behavioral control, coercive control, and overprotection, and consists of a total of 15 dimensions [30, 33]. Nurturance is described as how responsive and supportive a parent is to a child's individuality and their needs. It includes the dimensions of responsiveness, autonomy support, social reward, and involvement [30]. Consistency, organization, scaffolding, and inconsistent discipline form the second construct called structure, which describes parents' capacity to organize the environment around their child. Behavioral control, the third construct, is a controlling style which is characterized by managing and supervising activities, while not over-controlling the child. The behavioral control construct consists of three dimensions; monitoring, maturity demands, and non-intrusive discipline. Coercive control, the fourth construct, is defined as dominating, discouraging, and pressuring the child's independence. It is measured by three dimensions: authoritarian control, physical punishment, and psychological control. Finally, the fifth construct, called overprotection, interferes with the child's independence by, for example, extensive involvement, such as not letting the child be involved in activities that may be dangerous.

The response categories in the CGPQ are on a five-point Likert scale, from strongly disagree (1) to strongly agree (5). Responses of inconsistent discipline items in the structure construct need to be reversed to fit with the other dimensions. Study 2 contained 29 items from the 69-item CGPQ, and still assessed all five higher-order constructs, but not all the dimensions (see S1 Table).

**Other measurements.** The questionnaires of Study 1 and Study 2 contained similar questions on the respondents' characteristics: the respondent's relationship with the child, the age of the respondent and the child, and the respondent's highest educational level, occupational status, and marital status. In addition, the respondent reported how many days per week the child usually attended preschool. The gender of the child was only elicited in Study 2.

## Statistical analyses

The characteristics of the study samples are described by frequencies, means, and standard deviations.

We present three versions of the CGPQ scale, all of which include the five higher-order constructs: Version 1––the "original" scale, with 69 items; Version 2––a scale with 29 items (reduced from the 69-item version); and Version 3––a scale with 22 items (reduced from the 69-item version).

The data from Study 1 (n = 173) were used for the first analyses, and confirmatory factor analyses (CFAs) was carried out in order to reduce the number of items that would then be used in Study 2. The item reduction process followed the guidelines step by step: all the items with low communalities (defined as standardized loadings < .40) in the first stage of the CFAs

were removed, then the skewness of the items was checked, and the highly skewed items were removed (skewness over +2/-2). Item reduction was also based on criteria other than the CFA results. For instance, the dimension of physical punishment was removed due to the Finnish law forbidding physical punishment of children (Act on Child Custody and Right of Access 361/1983). In addition, items were removed or included on the basis of the criterion that each dimension should have at least two items representing the dimension in the reduced version. If the statistical analyses results showed three or four items for a dimension, we looked for the item(s) that had similar meanings/contents in the Finnish context, and then reduced the items so that only one/two items represented that meaning. This was done to further reduce the total number of items in the instrument. As the overprotection construct is understudied in relation to children's EBRBs and weight [30], this construct was not reduced by this "meaning criteria" as much as the others. If exclusion/inclusion was based on premises other than statistical ones, each item and its interpretation was discussed with other experts and parents of young children. The result of the reduction process, the statistical analyses and their results, as well as the reasons for excluding/including items, are presented in detail in S1 Table: Result from the statistical analyses for the reduction process, and reasons others than statistical for excluding/including items in the reduced 29-item version. A total of 40 items were removed, leaving a 29-item version for Study 2.

The information for the 69 items was hence only available in the Study 1 data. After some statistical evaluations on the importance of each item (detailed below) another reduced alternative version (Option 3) was formed, containing 22 items. Data from Study 2, with a higher number of respondents (parents of 805 children), were used to confirm the results for Option 2 and Option 3. For all three options we evaluated the internal consistency of the constructs and items using Cronbach's alpha, and we compared the reduced version scales with 29 and 22 items with the original scale with 69 items. Intra-class correlation (ICC) was explored and used to evaluate the reliability and homogeneity of the results of the two CGPQ versions, separately for Study 1 and Study 2 respondents. We evaluated the convergent validity by the average variance extracted (AVE) for each latent factor. AVE over 0.50 is recommended for each construct, indicating a good convergent validity [39].

In order to evaluate the construct validity of the reduced versions (29 and 22 items) and check the importance of the items for the scales, a combined assessment of the CFA and items response theory (IRT) results was used. The CFA was used to check the scale's factor structures. Estimations of the CFAs were carried out using a robust unweighted least squares method. The goodness-of-fit of the CFA models were assessed using the following measures: the comparative fit index (CFI), non-normed fit index (NNFI), root mean square error (RMSEA), standardized mean square residual (SRMR), goodness-of-fit index (GFI), and adjusted goodness-of-fit index (AGFI) [40]. IRT was used to assess each item in the scale and to measure the item importance and how each item could discriminate the participant response related to the scale. For this purpose, we present the item discrimination parameter, which describes how an item can differentiate between individuals [41]. We only considered excluding items with very low discrimination power (at <0.20).

We used the Lavaan [42] and mirt packages [43] in R for the CFA and IRT, respectively, and SPSS statistical software (version 24.0) for the other statistical analyses. We adopted a 5% statistical significance level for all tests.

## Results

Table 1 shows the descriptive characteristics of the study samples. In Study 1, 92% of the respondents were mothers, versus 87% in Study 2. In general, the educational level of the

**Table 1. Descriptive characteristics of the two study samples.**

| | | Study 1, % (N = 173) | | Study 2, % (N = 806) | |
|---|---|---|---|---|---|
| Respondent | mother | 92 (N = 159) | | 88 (N = 698) | |
| | father, grandparent, foster parent | 8 (N = 14) | | 13 (N = 99) | |
| Educational level | low | 16 (N = 28) | | 29 (N = 232) | |
| | middle | 33 (N = 57) | | 41 (N = 327) | |
| | high | 42 (N = 73) | | 29 (N = 233) | |
| | other, would rather not answer | 8 (N = 15) | | | |
| Marital status | married/ registered partnership/ cohabiting | 88 (N = 153) | | 90 (N = 723) | |
| | other | 12 (N = 20) | | 10 (N = 72) | |
| Occupation | full time work (incl shift work) | 48 (N = 83) | | 80 (N = 634) | |
| | part time/other | 52 (N = 90) | | 20 (N = 160) | |
| Gender of child | girl | NA | | 48 (N = 391) | |
| | boy | NA | | 52 (N = 415) | |
| Preschool/early education | full time (at least 4 days/week) | 70 (N = 121) | | 82 (N = 657) | |
| | part time/other | 30 (N = 52) | | 18 (N = 141) | |
| | | Mean (SD) | range | Mean (SD) | range |
| Age of respondent | | 36.8 (6.0) | 22 to 72 | 35.9 (4.9) | 23 to 53 |
| Age of child | | 4.3 (1.1) | 2 to 7 | 4.3 (1.0) | 2 to 7 |

*SD standard deviation

respondents in Study 1 was higher, and less of them worked full time. Respondents in Study 1 had children who less often attended preschool four to five days per week, than those who participated in Study 2. In Study 1, the age range of the respondents was 22–72 years, whereas in Study 2 the age range was 23–53 years. The mean age of the respondents was similar in both studies, 37 (SD 6) in Study 1 and 36 years (SD 4.9) in Study 2. The mean age of the children was about 4.3 years old in Studies 1 and 2, with SD 1.1 and SD 1.0, respectively.

Table 2 presents Cronbach's alpha for the five constructs in Study 1 and Study 2. For Study 1, Cronbach's alpha of the five constructs for all three options is presented: Option 1; 69 items, Option 2; 29 items, and Option 3; 22 items. For Study, 2 Option 2 and Option 3 are also presented. In Study 1, higher Cronbach's alphas were generally found in Option 1, which was

**Table 2. Internal consistency–Cronbach's alpha per construct considering three versions; all 69 items, 29 items, and 22 items (Study 1 and Study 2 data).**

| Items | Option 1 (all items) | Option 2 (29 items) | Option 3 (22 items) |
|---|---|---|---|
| **Study 1 data** | | | |
| Behavior control | 0.74 | 0.58 | 0.53 |
| Coercive control | 0.65 | 0.54 | 0.59 |
| Nurturance | 0.76 | 0.63 | 0.58 |
| Overprotection | 0.59 | 0.60 | 0.60 |
| Structure | 0.74 | 0.67 | 0.64 |
| **Study 2 data** | | | |
| Behavior control | | 0.62 | 0.61 |
| Coercive control | | 0.46 | 0.60 |
| Nurturance | | 0.67 | 0.63 |
| Overprotection | | 0.58 | 0.58 |
| Structure | | 0.63 | 0.59 |

**Table 3. Intra-class correlation between versions created using a different number of items for Study 1 and Study 2 data.**

| | Intraclass correlation | | |
|---|---|---|---|
| | 69 item x 29 items | 69 item x 22 items | 29 item x 22 items |
| **Study 1 data** | | | |
| Behavioral control (mean) | 0.86 | 0.79 | 0.95 |
| Coercive control (mean) | 0.71 | 0.51 | 0.76 |
| Nurturance (mean) | 0.87 | 0.84 | 0.99 |
| Overprotection (mean) | 0.94 | 0.94 | 1.00 |
| Structure (mean) | 0.78 | 0.61 | 0.88 |
| **Study 2 data** | | | |
| Behavioral control (mean) | | | 0.95 |
| Coercive control (mean) | | | 0.76 |
| Nurturance (mean) | | | 0.99 |
| Overprotection (mean) | | | 1.00 |
| Structure (mean | | | 0.87 |

expected, as there were many more items in this option. The results varied between 0.59 and 0.76 for the five constructs in Option 1, between 0.54 and 0.67 in Option 2, and between 0.53 and 0.64 in Option 3. In Study 2, Cronbach's Alpha was assessed for Option 2 and Option 3. In Option 2, the lowest value was found for coercive control (0.46) and the highest values for nurturance (0.67). In Option 3, higher Cronbach's alphas were found, ranging from 0.58 to 0.63.

Table 3 presents the intraclass correlations two by two for each of the five constructs. The reliability and homogeneity are reported for Study 1 first, and after this for the Study 2 data. In the Study 1 data, the 69-item version showed higher concordance with the 29-item version than with the 22-item version for all constructs, but high concordance was also found between the 22-item and the 29-item version (Table 3). The lowest intraclass correlation in Study 1, 0.51, was found between Option 1 (69 items) and Option 3 (22 items) for the construct of coercive control. The intra-class correlation was 1.00 for overprotection in Option 2 and Option 3, because the same items were included in both options. In Study 2, the higher concordance between Option 2 and Option 3 was confirmed by even higher values. The intra-class correlations between these options varied between 0.76 for the construct of coercive control and 1.00 for the construct of overprotection, showing a high concordance between the scales for all constructs.

Further CFAs, presented as standards estimations, and the ITR, reported as discrimination parameters, were conducted in the Study 1 sample in order to evaluate the factorial structure of the scale and the influence of each item on the entire scale (Table 4). The detailed results of the reduction process of 69-items into the 29-item scale used in Study 2, are presented in S1 Table; Results from the statistical analyses for the reduction process, and reasons others than statistical for excluding/including items in the reduced 29-item version. The results in Table 4 for Study 2 shows that the standard estimations derived from the CFAs show low estimations for six items (0.16–0.21), highlighting the low influence on the scale and suggesting the exclusion of these items. The discrimination parameters in the IRT analysis further confirmed the exclusion of one of those six items with a low value of 0.20. Another item with a low parameter of 0.19 was also excluded on the basis of the IRT result. One of the six items showed a low standard estimation of 0.18 in the CFA and a modest low parameter value of 0.41, and was thus excluded. This meant that, in all seven exclusions, the scale ended up with 22 items. After the

**Table 4. Estimations and results of Confirmatory Factor Analysis (CFA) and Item Response Theory (IRT) in Study 1 data.**

| Item number in CGPQ and item | CFA | | IRT | Exclusion criteria | |
|---|---|---|---|---|---|
| | Standard estimation (29 items) | Standard estimation, after exclusions (22 items) | Discrimination parameter (Data with 29 items) | based on CFA | based on IRT |
| **Behavioral control** | | | | | |
| 13. I expect my child to follow our family rules | 0.23 | 0.20 | 0.35 | | |
| 38. I pay close attention to where my child is | 0.58 | 0.62 | 1.11 | | |
| 52. I make sure I know where my child is at all times | 0.69 | 0.76 | 4.84 | | |
| 62. I teach my child to follow rules | 0.28 | 0.24 | 0.30 | | |
| 68. I make sure that my child understands what I expect of him/her | **0.21** | | **0.20** | item 68 | item 68 |
| **Coercive control** | | | | | |
| 4. When my child does something that is not allowed, I do not talk to him/her for a while | **0.21** | | 0.48 | item 4 | |
| 10. I want my child to always obey me | 0.66 | 0.70 | 1.02 | | |
| 23. I place a lot of emphasis on my child's obedience | 0.56 | 0.52 | 1.15 | | |
| 30. I make my child feel bad when he/she does not meet my expectations | **0.18** | | 0.41 | item 30 | item 30 |
| **Nurturance** | | | | | |
| 7. I know exactly when things are not going very well for my child | 0.26 | 0.29 | 1.25 | | |
| 17. I let my child make his/her own choices as long as they are safe | 0.25 | 0.25 | 1.01 | | |
| 21. I say something nice to my child as a reward for good behavior | 0.21 | 0.22 | 0.40 | | |
| 26. When my child does his/her best, I praise him/her | **0.16** | | 1.31 | item 26 | |
| 40. I spend a lot of time with my child | 0.36 | 0.31 | 0.57 | | |
| 44. I easily find a way to make time for my child | 0.29 | 0.28 | 0.35 | | |
| 59. I encourage my child to approach things his/her own way, even if it means more work for me | 0.34 | 0.33 | 1.02 | | |
| 67. I know exactly when my child has difficulty with something | 0.33 | 0.36 | 0.97 | | |
| **Overprotection** | | | | | |
| 12. Every free minute I have, I spend with my child | 0.54 | 0.52 | 0.50 | | |
| 16. I always help my child with everything he/she does | 0.69 | 0.67 | 0.81 | | |
| 41. When my child cannot find something, I stop what I am doing to find it before he/she gets too upset | 0.55 | 0.58 | 0.85 | | |
| 47. I do not let my child get involved in activities or tasks where he/she might get hurt | 0.35 | 0.38 | 0.49 | | |
| 53. I carefully plan my child's day so that he/she has enough activities to keep him/her busy | 0.37 | 0.35 | 0.60 | | |
| **Structure** | | | | | |
| 2. When I tell my child I will do something, I do it | **0.19** | | 0.43 | item 2 | |
| 8. I use clear and consistent messages when I tell my child to do something | **0.20** | | 0.74 | item 8 | |
| 28. I try to make sure that my child has a regular schedule from day to day | 0.51 | 0.32 | 0.21 | | |
| 29. I put time and energy into helping my child, when he/she needs it | 0.33 | 0.32 | 1.89 | | |
| 35. I organize my child's week so that it follows a regular, predictable pattern | 0.58 | | **0.19** | | item 35 |
| 57. I make sure my child is at activities on time | 0.27 | 0.30 | 3.68 | | |
| 63. When my child has a problem, I help him/her figure out what to do about it | 0.26 | 0.29 | 2.77 | | |

**Table 5. Results for the average variance extracted (AVE) for each factor.**

| Construct | Study 1 | Study2 |
|---|---|---|
| Behavioral control | 0.70 | 0.80 |
| Coercive control | 0.58 | 0.88 |
| Nurturance | 0.80 | 0.85 |
| Overprotection | 0.76 | 0.82 |
| Structure | 0.69 | 0.82 |

exclusion of these seven items (two suggested by CFA and IRT, four suggested by CFA and one suggested by IRT) and a rerun of the CFA, the result showed that the standard estimations for many items increased and that the lowest standard estimation in the model was now 0.22 and the highest 0.76. The AVE for ACF estimated in Study 1, showing value over 0.5 for all constructs, indicating a good convergent validity, and that no other item needed to be deleted (Table 5). The AVEs for the constructs estimated in Study 2 were even higher, with all values over 0.80, demonstrating excellent results. The standard estimation obtained for Study 2 are presented in Fig 1, in which all estimated value with p<0.001.

Table 6 shows the results of the goodness-of-fit measures for all the models evaluated in the Study 1 and Study 2 samples through CFA. In the Study 1 sample, goodness-of-fit was checked for all three options. None of the three options met the criteria for all the indicators; the model with better goodness-of-fit was Option 3, in which three indicators met the adopted criteria. The CFA of Study 2 data (Table 6) showed that most criteria for a satisfactory goodness-of-fit were met. However, in Option 2, the TLI (0.91) and the NNFI (0.91), did not fulfill the criteria of ≥0.95. In Option 3, all the criteria for the assessments were met, which means that the model with 22 items was the better model, e.g., a scale with a better factor structure.

## Discussion

The main aim of this study was to evaluate the factor structure of the reduced versions of the validated 69-item CGPQ for young children. The psychometrics properties of the reduced versions were examined by the item response theory, the CFA to evaluate the factor structure, internal consistency, and internal validity. Initially, a pilot study was used (Study 1) to pre-assess the 69-item version, followed by the DAGIS survey (Study 2), which used a larger number of participants to validate and assess the reliability of the reduced versions. Our analysis of the reduced versions (Option 2 with 29-items and Option 3 with 22-items) revealed that both had satisfactory goodness-of-fit, and both reduced versions had good concordance with the 69-item version.

Reliability was assessed as the internal consistency of all three versions in the Study 1 data, and for the two reduced versions in the Study 2 data. As expected, the highest Cronbach's alphas were found for the five separate constructs in the 69-item version, as this version included a higher number of items per construct. However, in general, Cronbach's Alphas for the 69-item version were lower than those in the in the study of van der Horst and Sleddens, which evaluated a higher number of participants using the same version of the CGPQ [33]. In their study, the coefficients varied between 0.65 and 0.91, whereas in Study 1 the coefficients in the constructs varied between 0.59 and 0.76. The mean age of the children in the study of van der Horst and Sleddens was two years, whereas in our study it was 4.3 years. This might have affected the parents' responses. However, using the original CGPQ for 5-13-years-old [33] was not a valid alternative, as the children in Study 1 were too young for this questionnaire. In Study 2, which had a higher number of participants (n = 805), the 22-item version showed

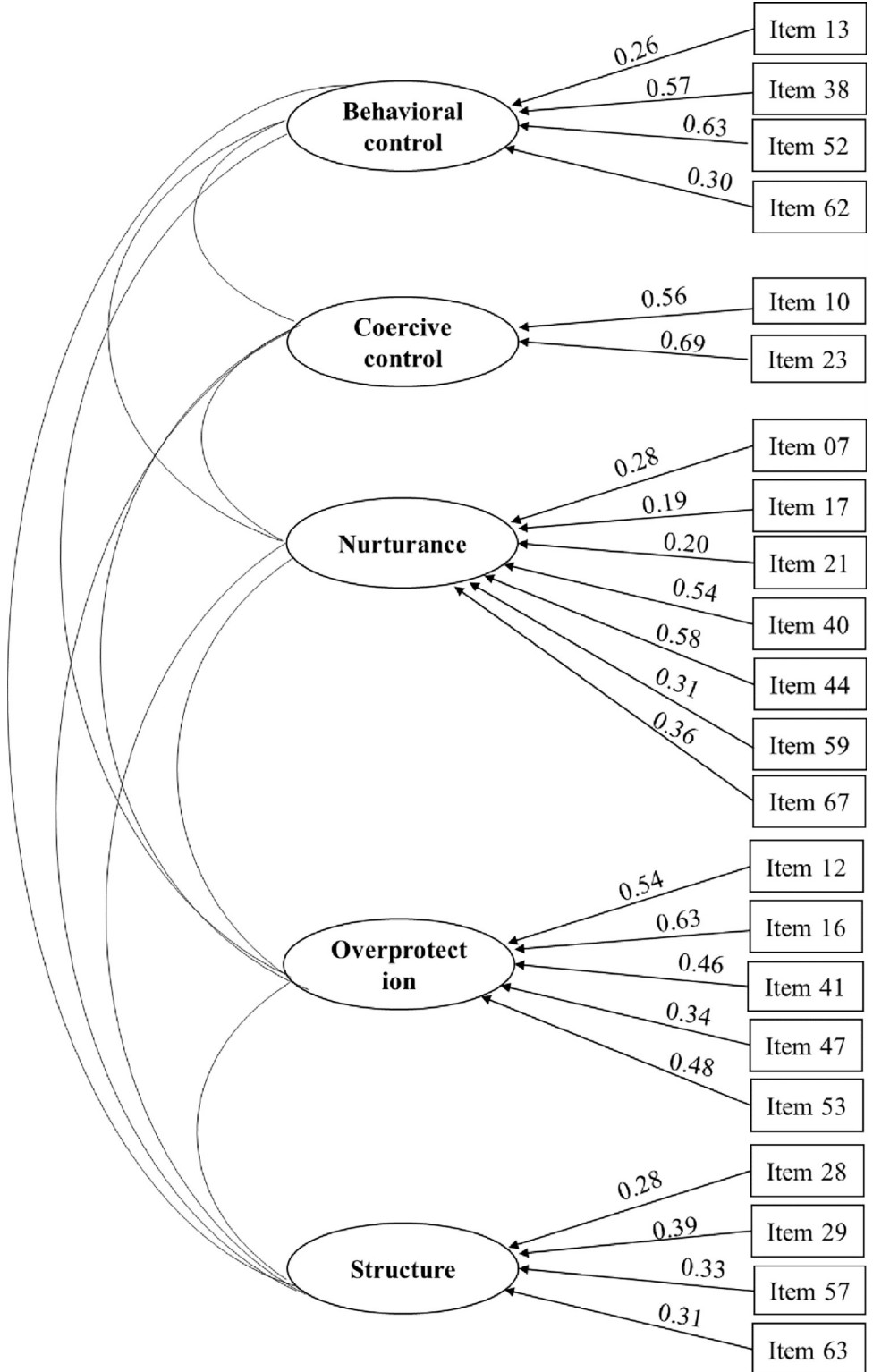

**Fig 1. Standard estimation of Confirmatory Factor Analysis (CFA) for the 22 items (Study 2).** All estimates p<0.001.

**Table 6. Detailed goodness-of-fit of models for CFA of three different versions evaluated in Study 1 and Study 2 data.**

| | Option 1 (all items used in Study 1) | Option 2 (29 items used in Study 2) | Option 3 (22 items used in Study 2, after exclusions by CFA and IRT) | Criteria (ref) |
|---|---|---|---|---|
| **Study 1 data** | | | | |
| Comparative Fit Index (CFI) | 0.59 | 0.76 | 0.85 | CFI ≥.90 |
| Tucker-Lewis Index (TLI) | 0.58 | 0.74 | 0.83 | NNFI ≥0.95 |
| NNFI | 0.58 | 0.74 | 0.83 | NNFI ≥0.95 |
| RMSEA | 0.08 | 0.08 | **0.04** | RMSEA <0.08 |
| SRMR | 0.11 | 0.10 | 0.09 | SRMR <0.08 |
| GFI | 0.84 | 0.93 | **0.96** | GFI ≥0.95 |
| AGFI | 0.83 | **0.92** | **0.95** | AGFI ≥0.90 |
| **Study 2 data** | | | | |
| Comparative Fit Index (CFI) | | **0.92** | **0.96** | CFI ≥.90 |
| Tucker-Lewis Index (TLI) | | 0.91 | **0.96** | NNFI ≥0.95 |
| NNFI | | 0.91 | **0.96** | NNFI ≥0.95 |
| RMSEA | | **0.07** | **0.03** | RMSEA <0.08 |
| SRMR | | **0.07** | **0.06** | SRMR <0.08 |
| GFI | | **0.96** | 0.98 | GFI ≥0.95 |
| AGFI | | **0.95** | 0.97 | AGFI ≥0.90 |

slightly higher coefficients for the higher-order constructs–between 0.58 and 0.63. It should be noted that each construct included less items than the original CGPQ, and therefore the result could be interpreted as satisfactory [44]. Also, we had higher coefficients than in a study where the same CGPQ was modified into 51 items [45]. Respondents in this previous study were 11-15-years-old and reported their mother's and father's parenting. Testing the five constructs resulted in 0.52–0.61 estimates for mothers and 0.42–0.62 for fathers [45]. Study 1 and Study 2 data both showed good intraclass correlation coefficients (ICC>0.70) for each separate higher-order construct [38].

Construct validity was verified by testing the structural validity using CFA and IRT. The low CFA estimates of the 29-item version showed that 6 out of the 29 items should be excluded. The IRT tests, used for testing the discriminating of the items [41], also confirmed that additional items should be excluded from the scale. A new CFA with 22 items showed an excellent goodness of fit, the model met all the CFA criteria [40], even though some items had low influence on the scale (low estimate). The CGPQ was originally validated in Dutch and English [30]. It has been recommended that construct validation should also include cross-cultural validation [46], by methods such as translations and back-translations, expert revision of the instrument, or pre-testing of the instrument. In our study, cross-cultural validity for the 69-item version was examined, as the version was translated and back-translated by native English-speakers, the developer (EB) of the CGPQ was involved during the process, experts revised it, and the comprehensibility of the Finnish and Swedish translated versions was tested on parents of young children. Still, our cross-cultural validation was conducted between countries which are all part of the Western societies. Testing an instrument in different cultural contexts has been discussed [47, 48]. A cross-cultural validity testing enhances the validity of the instrument and it will strengthen the generalizability of study results.

This study has limitations that should be acknowledged. The Study sample 1 was relatively small (n = 173) for conducting a proper CFA to reduce the 69-item version. As a rule of thumb, a higher number of participants would be needed to obtain better estimations [48]. However, the internal consistency of the high-order constructs showed higher Cronbach's Alpha values. One can speculate whether the constructs assessed parenting comprehensively enough after the items were reduced. So, even though the construct validity in the 29-item, and especially in the 22-item version was good, further studies could further validate the content. Another limitation of the study was that we were unable to evaluate the 69-item version of the CGPQ in Study 2. In addition to these limitations, Study 1 did not include a question about the gender of the child, which could have been useful background information for the study.

The relatively high number of participants in Study 2 could be seen as a strength of this study. This enabled comprehensive reliability and construct validity tests of the 29- and 22-item versions. After running the statistical analyses, the 29- and 22-item version in Study 2, especially the 22-item version, showed that the reduced version is a valid instrument for use in the Finnish context, with satisfactory goodness-of-fit and high concordance with the original 69-item scale. The reduced 22-item version could be a less burdensome instrument for upcoming studies. In addition, the new knowledge derived could be a valuable contribution to studies examining parenting, children's EBRBs, or the risk of overweight and obesity. As the reduced instrument includes all five higher-order constructs, it is an useful feasible instrument in testing possible moderating effects of parenting on associations between parenting practices and children's EBRB's, in accordance with the original conceptual model of parenting style and discussion brought up in the literature [8, 12, 19–21]. Still, when comparing previous studies, our study, and the study in which it was used on adolescents [45], it seems that the instrument works better when the respondents are parents, as the reliability was generally lower in the adolescents' study [27, 31, 33, 45]. Further studies could focus on adjusting the instrument so it is suitable for adolescents and assesses perceived parenting in a useful and meaningful way. However, the result of this study already broadens the target group for the instrument. The 69-item instrument was originally developed and tested for parents of 1-4-years-old, whereas the current study included parents of slightly older children, 3-6-years-olds. In future studies, the five higher-order parenting constructs could be used separately, which enables the differentiation between positive and negative control (behavioral control versus coercive control and overprotection), or the five higher-order constructs could be clustered in order to reflect overall parenting style.

## Conclusion

This study showed that the shortened 22- and 29-item versions are comparable with the 69-item CGPQ. The psychometrics of the 22-item version showed stronger results when describing the five higher-order constructs. However, as several items were dropped, one can conclude that the 22-item reduced version of the 69-item questionnaire assesses the five higher-order constructs well, but it does not cover all dimensions in the original instrument. To conclude, the 22-item CGPQ showed good construct validity, which makes it a short, valid, and feasible instrument for assessing parenting, and it covers the constructs of both positive and negative control which is valuable in studies aiming to examine children's EBRBs and weight.

## Supporting information

**S1 Table. Result from the statistical analyses for the reduction process, and reasons others than statistical for excluding/including items in the reduced 29-item version.**
(DOCX)

## Acknowledgments

The authors thank the families who participated in the WEB survey (Study 1); the preschools, the preschool personnel, and the families for their participation in the DAGIS study; and the research staff for data collection. The authors also thank the collaborating partners of the DAGIS study for providing assistance in designing the study.

## Author Contributions

**Conceptualization:** Carola Ray, Ester van der Borgh-Sleddens, Mona Bjelland.

**Formal analysis:** Carola Ray, Rejane Augusta de Oliveira Figueiredo.

**Funding acquisition:** Eva Roos.

**Methodology:** Carola Ray, Ester van der Borgh-Sleddens, Rejane Augusta de Oliveira Figueiredo.

**Project administration:** Carola Ray, Eva Roos.

**Visualization:** Carola Ray, Rejane Augusta de Oliveira Figueiredo.

**Writing – original draft:** Carola Ray.

**Writing – review & editing:** Ester van der Borgh-Sleddens, Rejane Augusta de Oliveira Figueiredo, Jessica Gubbels, Mona Bjelland, Eva Roos.

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
