## [Decision Letter · Decision Letter 0]

7 Mar 2022

PONE-D-21-32971Psychometric properties of the item-reduced version of the Comprehensive General Parenting Questionnaire for caregivers of preschoolers in a Finnish contextPLOS ONE

Dear Dr. Ray,

Thank you for submitting your manuscript to PLOS ONE. After careful consideration, we feel that it has merit but does not fully meet PLOS ONE’s publication criteria as it currently stands. Therefore, we invite you to submit a revised version of the manuscript that addresses the points raised during the review process.

Apologies for the long time spent processing this first version of your paper. Unfortunately, it was really difficult to locate suitable (and available) reviewers for this paper. Anyway, I got a first review report, with which (given the situation) I will go straightforward to you, in order to ask you for some (minor) revisions, in order to reconsider the acceptability of the paper. Although most of these changes are rather miscellaneous, please make sure to address all the comments provided by the referee, responding to them in the rebuttal letter.

We look forward to receiving your revised manuscript.

Kind regards,

Sergio A. Useche, Ph.D.

Academic Editor

PLOS ONE

Journal Requirements:

The DAGIS project and this study was financially supported by Folkhälsan Research Center, University of Helsinki, The Ministry of Education and Culture in Finland, The Ministry of Social Affairs and Health, The Academy of Finland (Grant: 285439, 315816), the Päivikki and Sakari Sohlberg foundation, and the Medicinska Föreningen Liv och Hälsa.

The DAGIS project and this study was financially supported by Folkhälsan Research Center, University of Helsinki, The Ministry of Education and Culture in Finland (ER), The Ministry of Social Affairs and Health (ER), The Academy of Finland (Grant: 285439, 315816) (ER), the Päivikki and Sakari Sohlberg foundation (ER), and the Medicinska Föreningen Liv och Hälsa (ER). The funders had no role in study design, data collection and analysis, decision to publish, or preparation of the manuscript.  

Reviewers' comments:

Reviewer's Responses to Questions

**Comments to the Author**

1. Is the manuscript technically sound, and do the data support the conclusions?

Reviewer #1: Partly

2. Has the statistical analysis been performed appropriately and rigorously? 

Reviewer #1: Yes

3. Have the authors made all data underlying the findings in their manuscript fully available?

Reviewer #1: Yes

4. Is the manuscript presented in an intelligible fashion and written in standard English?

Reviewer #1: Yes

5. Review Comments to the Author

Reviewer #1: This study assesses the factorial structure of the reduced item version of the Comprehensive General Parenting Questionnaire (CGPQ), an instrument that evaluates five general parenting constructs. The manuscript presents important practical implications, due to the need of assessing the validity of this type of questionnaires, considering the facility of their administration in comparison with more extended versions.

The introduction adequately presents the previous studies performed through different questionnaires framed within this field of study. However, I believe that the references are scarce, as the issue needs to be contextualized and the study requires an adequate theoretical framework. Therefore, I suggest a further effort to complement this section.

The methodology is adequately presented, but I suggest including some table or figure clarifying the dimensions that should be obtained from the questionnaire. Likewise, it would also be convenient to explain the psychometric properties of each item through a table, as well as including a figure explaining the correspondence between the items and the dimensions of the questionnaire (the following article may be useful for this purpose: https://doi.org/10.1016/j.trf.2018.08.003). Moreover, the average variance extracted (AVE) and the adjustment indexes of the scale must be included.

The discussion summarizes the main findings of the study, but this section must also be complemented with other studies that may serve to contrast the results. In addition, the practical implications of the research could be included in the discussion/conclusion.

For what concerns the style and writing of the manuscript, they are adequate.

6. PLOS authors have the option to publish the peer review history of their article (what does this mean?). If published, this will include your full peer review and any attached files.

Reviewer #1: No

---

## [Author Response · Author response to Decision Letter 0]

13 May 2022

Dear Dr. Ray,

Thank you for submitting your manuscript to PLOS ONE. After careful consideration, we feel that it has merit but does not fully meet PLOS ONE’s publication criteria as it currently stands. Therefore, we invite you to submit a revised version of the manuscript that addresses the points raised during the review process.

Apologies for the long time spent processing this first version of your paper. Unfortunately, it was really difficult to locate suitable (and available) reviewers for this paper. Anyway, I got a first review report, with which (given the situation) I will go straightforward to you, in order to ask you for some (minor) revisions, in order to reconsider the acceptability of the paper. 

Although most of these changes are rather miscellaneous, please make sure to address all the comments provided by the referee, responding to them in the rebuttal letter.

We look forward to receiving your revised manuscript.

Kind regards,

Sergio A. Useche, Ph.D.

Academic Editor

PLOS ONE

Journal Requirements:

Answer: We have named the files as instructed

Answer: We have clarified the details regarding participation in studies 1 and 2 in the manuscript (lines 128-136). In study 1, the respondents were informed as written text in the beginning of the online questionnaire that by answering they were agreeing to participate in the study. We present the information about the written consent in study 2, as well. We have also corrected the details in the online submission.

The DAGIS project and this study was financially supported by Folkhälsan Research Center, University of Helsinki, The Ministry of Education and Culture in Finland, The Ministry of Social Affairs and Health, The Academy of Finland (Grant: 285439, 315816), the Päivikki and Sakari Sohlberg foundation, and the Medicinska Föreningen Liv och Hälsa.

The DAGIS project and this study was financially supported by Folkhälsan Research Center, University of Helsinki, The Ministry of Education and Culture in Finland (ER), The Ministry of Social Affairs and Health (ER), The Academy of Finland (Grant: 285439, 315816) (ER), the Päivikki and Sakari Sohlberg foundation (ER), and the Medicinska Föreningen Liv och Hälsa (ER). The funders had no role in study design, data collection and analysis, decision to publish, or preparation of the manuscript. 

Answer: Thank you for this comment. We have excluded this information from the manuscript. In the cover letter we have included what should be stated about the funding. 

Answer: We have described how data is available in the cover letter. 

Answer: We have included captions about our Supplementary Table at the end of the manuscript. In-text citations are also updated.

Answer: We are sorry, but we do not understand the comments about citing retracted papers?

We have added references in the manuscript. These references are mentioned in the rebuttal letter when discussing the comments with the reviewer. 

Reviewers' comments:

Reviewer #1: This study assesses the factorial structure of the reduced item version of the Comprehensive General Parenting Questionnaire (CGPQ), an instrument that evaluates five general parenting constructs. The manuscript presents important practical implications, due to the need of assessing the validity of this type of questionnaires, considering the facility of their administration in comparison with more extended versions.

The introduction adequately presents the previous studies performed through different questionnaires framed within this field of study. 1)However, I believe that the references are scarce, as the issue needs to be contextualized and the study requires an adequate theoretical framework. Therefore, I suggest a further effort to complement this section.

Answer: Thank you for the comment. We have complemented several sections in the introduction to conceptualize the issue and the larger theoretical framework. We added references related to these issues throughout the introduction. Please see the changes in lines 62-68, 79-82, 87-88, and 92-95. 

The added references in the introduction are:

Howe AS, Heath A-LM, Lawrence J, Galland BC, Gray AR, Taylor BJ, et al. (2017). Parenting style and family type, but not child temperament, are associated with television viewing time in children at two years of age. PLoS ONE 12(12): e0188558. https://doi.org/10.1371/journal.pone.0188558

Van der Geest KE, Mérelle SYM, Rodenburg G, Van de Mheen, Renderset CM. Cross-sectional associations between maternal parenting styles, physical activity and screen sedentary time in children. BMC Public Health (2017) 17:753 DOI 10.1186/s12889-017-4784-8

Burnett AJ, Lamb KE, Spence AC, Lacy KE, Worsley A. Parenting style as a predictor of dietary score change in children from 4 to 14 years of age. Findings from the Longitudinal Study of Australian Children. Public Health Nutr. 2021 Dec;24(18):6058-6066. doi: 10.1017/S1368980021003062. Epub 2021 Jul 23. PMID: 34296665.

Philips N, Sioen I, Michels N, Sleddens E, De Henauw S. The influence of parenting style on health related behavior of children: findings from the ChiBS study. International Journal of Behavioral Nutrition and Physical Activity 2014, 11:95 http://www.ijbnpa.org/content/11/1/95

Gerards SMPL, Niermann C, Gevers DWM, Eussen N, Kremers SPJ. Context matters! The relationship between mother-reported family nutrition climate, general parenting, food parenting practices and children’s BMI. BMC Public Health (2016) 16:1018 DOI 10.1186/s12889-016-3683-8.

The methodology is adequately presented, but I suggest 2) including some table or figure clarifying the dimensions that should be obtained from the questionnaire. 

Answer: In the S1 Table (Supplementary Information) we present in the first column each of the five construct and their included items. The second column includes all the dimensions. We prefer to have this table as a Supplementary file, as it is very large, including the psychometrics of each item in the 69-item scale. We have in the results section added a sentence about the S1 Table (lines 280-283), and in the end of the manuscript we added a notice about that the manuscript includes Supplementary Information. In addition, the dimensions are presented in both Table 5 and Figure 1 in the main manuscript.

Likewise, it would also be convenient to 3) explain the psychometric properties of each item through a table, as well as including a figure explaining the correspondence between the items and the dimensions of the questionnaire (the following article may be useful for this purpose: https://doi.org/10.1016/j.trf.2018.08.003).

Answer: Thank you for your suggestion. In our study the psychometric properties of the scale were examined by the item response theory, the AFC to evaluate the factor structure, internal consistency, and internal validity as described in the text (Added in the beginning of the discussion in lines 324-325. And as suggested, we present in line 340 a figure named Figure 1. Standard estimation of Confirmatory Factor Analysis (CFA) for the 22 items (Study 2). All estimates p<0.001. The figure is pointing out the correspondence between the items and each construct of the questionnaire.

4) Moreover, the average variance extracted (AVE) and the adjustment indexes of the scale must be included.

Answer: Thank you for this important suggestion. 

We have added the AVE and following changes have been done in the manuscript.

- We have in the statistical part presented that we will examine the AVE and we have added a reference as well, lines 217-219: “We evaluated the convergent validity by the average variance extracted (AVE) for each latent factor. AVE over 0.50 is recommended for each construct, indicating a good convergent validity [Hair J 2010].”

- Added reference: Hair J., Black W., Babin B., Anderson R. (2010). Multivariate Data Analysis, 7th Edn Upper Saddle River, NJ: Prentice-Hall, Inc. 

In addition, we calculated and included the AVE in the result section (Lines 293-297 and Table 5). 

- The adjustment indexes were presented in Table 6. "Detailed goodness-of-fit of models for CFA of three different versions evaluated in Study 1 and Study 2 data".

- New text in the document: “The AVE for ACF estimated in Study 1, shown value over 0.5 for all constructs, indicating a good convergent validity, and that no other item needed to be deleted (Table 5). The AVEs for the constructs estimated in Study 2 were even higher, with all values over 0.80, demonstrating excellent results. The standard estimation obtained for Study 2 are presented in Fig 1, in which all estimated value with a significance p<0.001.”

The discussion summarizes the main findings of the study, but 5) this section must also be complemented with other studies that may serve to contrast the results. 

Answer: Overall, we have elaborated the discussion. We now refer to other studies, conducted with the CGPQ instrument, but also to other studies which have examined psychometric properties of an instrument.

The added sections about other studies are in lines 346-349, 364-368.

Added references:

Gevers DW, van Assema P, Sleddens EF, de Vries NK, Kremers SP. Associations between general parenting, restrictive snacking rules, and adolescent's snack intake. The roles of fathers and mothers and interparental congruence. Appetite. 2015 Apr;87:184-91. doi: 10.1016/j.appet.2014.12.220. Epub 2014 Dec 30. PMID: 25555538.

Bohman B, Rasmussen F, Ghaderi A. Development and psychometric evaluation of a context-based parental self-efficacy instrument for healthy dietary and physical activity behaviors in preschool children. International Journal of Behavioral Nutrition and Physical Activity (2016) 13:110 DOI 10.1186/s12966-016-0438-y.

6) In addition, the practical implications of the research could be included in the discussion/conclusion.

Answer: Thank you for the comment. We have added a section about practical implications and proposed some further studies. Please see lines 389-401. 

For what concerns the style and writing of the manuscript, they are adequate.

Thank you!

---

## [Decision Letter · Decision Letter 1]

20 Jun 2022

Psychometric properties of the item-reduced version of the Comprehensive General Parenting Questionnaire for caregivers of preschoolers in a Finnish context

PONE-D-21-32971R1

Dear Dr. Ray,

We’re pleased to inform you that your manuscript has been judged scientifically suitable for publication and will be formally accepted for publication once it meets all outstanding technical requirements.

Kind regards,

Sergio A. Useche, Ph.D.

Academic Editor

PLOS ONE

Reviewers' comments:

Reviewer's Responses to Questions

**Comments to the Author**

1. If the authors have adequately addressed your comments raised in a previous round of review and you feel that this manuscript is now acceptable for publication, you may indicate that here to bypass the “Comments to the Author” section, enter your conflict of interest statement in the “Confidential to Editor” section, and submit your "Accept" recommendation.

Reviewer #1: All comments have been addressed

2. Is the manuscript technically sound, and do the data support the conclusions?

Reviewer #1: Yes

3. Has the statistical analysis been performed appropriately and rigorously? 

Reviewer #1: Yes

4. Have the authors made all data underlying the findings in their manuscript fully available?

Reviewer #1: Yes

5. Is the manuscript presented in an intelligible fashion and written in standard English?

Reviewer #1: Yes

6. Review Comments to the Author

Reviewer #1: The authors have taken into account the suggestions I provided in my previous review, so I consider that the manuscript is suitable for publication.

7. PLOS authors have the option to publish the peer review history of their article (what does this mean?). If published, this will include your full peer review and any attached files.

Reviewer #1: No

---

## [Editor Report · Acceptance letter]

27 Jun 2022

PONE-D-21-32971R1 

Psychometric properties of the item-reduced version of the Comprehensive General Parenting Questionnaire for caregivers of preschoolers in a Finnish context 

Dear Dr. Ray:

I'm pleased to inform you that your manuscript has been deemed suitable for publication in PLOS ONE. Congratulations! Your manuscript is now with our production department. 

Kind regards, 

on behalf of

Dr. Sergio A. Useche 

Academic Editor

PLOS ONE